# Context-aware Cutmix Is All You Need for Universal Organ and Cancer Segmentation[⋆][⋆⋆]

Qin Zhou[1,2⋆], Peng Liu[1⋆], and Guoyan Zheng[1]

[1] Institute of Medical Robotics, School of Biomedical Engineering,
Shanghai Jiao Tong University, No. 800, Dongchuan Road, Shanghai 200240, China.
[2] Department of Computer Science and Engineering,
East China University of Science and Technology, Shanghai, 200237, China.
Correspondence: Guoyan Zheng (Email: guoyan.zheng@sjtu.edu.cn)

**Abstract.** Due to its important potential for various clinical applications, universal organ and cancer segmentation has attracted increasing attention recently. However, its performance is largely hindered due to issues such as (1) partial and noisy labels from different sources and (2) tremendously heterogeneous tumor cases. In this paper, we propose a novel partially supervised segmentation framework by introducing the $merge - max$ operation for hard mining among the unlabeled classes. Besides, to take full advantage of the expertly annotated tumor data, we design a novel context-aware CutMix scheme to dynamically perform tumor augmentation during training. We also introduce a useful data-cleaning strategy for self-training and adjust the nnU-Net framework for better efficiency. The average scores of organ DSC, organ NSD, tumor DSC and tumor NSD on the public validation set are 92.18%, 96.33%, 46.26% and 38.65%, respectively. And we achieve scores of 93.17% (organ DSC), 96.76% (organ NSD), 61.49% (tumor DSC) and 49.9% (tumor NSD) on the official test set. The average inference time is 13.95 seconds, the average maximum GPU memory is 2823 MB, and the average area under the GPU memory-time curve is 14112. Collectively, we ranked second among all submitted teams. Our code is available at https://github.com/luckieucas/FLARE23.

**Keywords:** Universal organ and cancer segmentation · Merge-max operation · Context-aware CutMix.

## 1 Introduction

Automatic multi-organ and cancer segmentation plays a vital role in computer-aided diagnosis and treatment planning. Recently, deep learning based methods have made remarkable progress in solving organ and tumor segmentation tasks. However, most of them focus only on one type of tumor (e.g., liver cancer,

---

[⋆] This study was partially supported by the Natural Science Foundation of China via project U20A20199 and 62201341.
[⋆⋆] Qin Zhou and Peng Liu contribute equally to this work.

kidney cancer). Pursuing a general and publicly available model for universal abdominal organ and cancer segmentation is rarely studied. Generally, multi-organ and cancer segmentation are faced with challenges from the following aspects: 1) Label inconsistency and partial annotations. Since the datasets are usually collected from different sources with varying purposes, not all the labels are annotated, and the same organ may be labeled as different indexes in different subsets; 2) Significant variations in the pan-cancer class. Due to different types of cancers, the non-rigidity of tumors and different disease progression across patients, the visual appearances of the pan-cancer class may vary dramatically among different individuals; 3) Label noise. Due to the requirement of expertise, noisy labels may occur in the forms of isolated points etc., especially in the annotations of pan-cancer class. This kind of noisy data can notoriously degrade the performance of NSD.

Many efforts have been devoted to solve the problems of label inconsistency and partial annotations. In PaNN [22], the average organ size distributions on the partially labeled datasets were constrained to resemble the prior statistics obtained from the fully labeled dataset. Another method was introduced in [17], where the non-overlapping characteristics between different organs was exploited to design the exclusion loss. Besides the prior-knowledege-based methods, co-training between two models with consistency constraints on soft pseudo labels [8], and multi-scale features learned in a pyramid-input and pyramid-output network [3] were both explored for partially supervised multi-organ segmentation. To allow effective training of the inconsistently labeled data, we introduce a $merge - max$ operation to perform hard mining on the unlabeled classes. Concretely, the labeled classes are constrained to be discriminative against each other and the hardest unlabeled class, where the hardest unlabeled class is identified as the index of the largest logits from the unlabeled classes.

To address the problem of tremendous intra-class variability (size, shape, positions and visual appearances etc.) of tumors, we propose a novel context-aware CutMix strategy to dynamically perform online tumor augmentation during training. Specifically, we locate all the tumor objects by finding each connected tumor regions, and cropping the tumor objects with neighboring regions to form a Context-aware Tumor Object (CTO) pool. These CTOs are then filtered and combined online with image cases not containing tumors for tumor augmentation during training.

To mitigate the effect of label noise, we introduce a novel data cleaning strategy to improve training data quality, which is proven to improve NSD segmentation performance without bells and whistles. (with only a small part of the unlabeled data, we can achieve equivalent performance compared to other methods that using all the unlabeled data. Besides, no post-processing is needed).

Our overall contributions can be summarized as follows:

– We propose a novel partially supervised segmentation framework by introducing the $merge - max$ operation for hard mining among the unlabeled classes.

– We design a novel context-aware CutMix scheme to dynamically perform tumor augmentation during training.
– We introduce a useful data-cleaning strategy for self-training and adjust the nnU-Net framework for better efficiency.

## 2   Method

In this section, we will elaborate on the training protocol of our method. Firstly, the partially labeled data is processed in a fine-resolution setting, and subsequently utilized to train a fine model for selecting the unlabeled data. Then a small part of the unlabeled data with confident pseudo labels are selected and combined with the partially labeled data to train a coarse model for final inference. During training, the $merge - max$ operation is introduced to perform hard mining on the unlabeled classes. Moreover, a novel context-aware CutMix is proposed for online tumor augmentation. Please note, data-cleaning is performed before coarse model training. We use the summation between Dice loss and cross-entropy loss, because compound loss functions have been proven to be robust in various medical image segmentation [11]. We also adapt the nnU-Net framework for better inference efficiency. Figure 1 illustrates our training protocol.

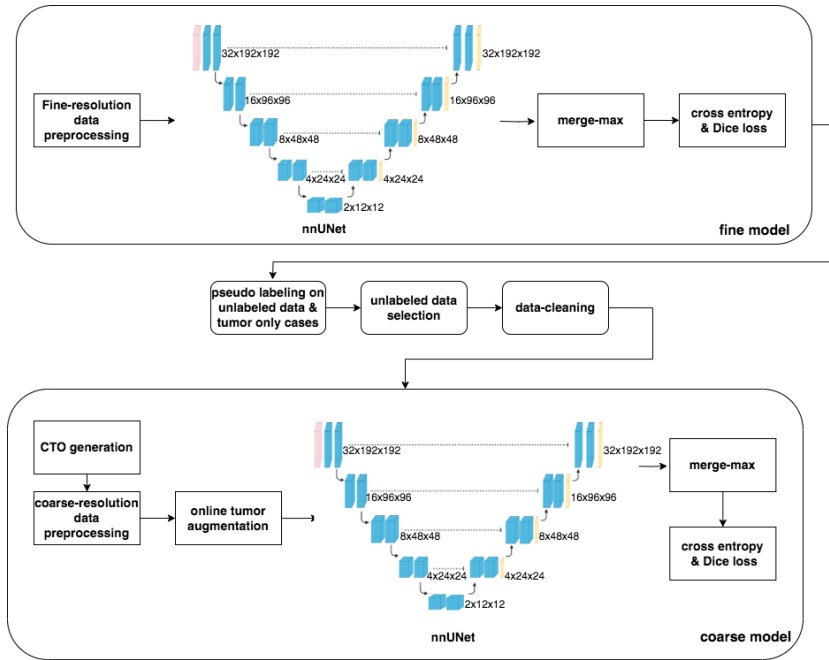

**Fig. 1.** A schematic illustration of our training protocol.

## 2.1   Preprocessing

We follow the standard preprocessing steps of nnU-Net to process the data before feeding into the network. For the fine-resolution and coarse-resolution preprocessing, the target spacing is (2.5,0.82,0.82) and (2.5,1.5,1.5) respectively.

## 2.2   Proposed Method

**Hard mining for partially supervised segmentation** As many organs or tumors are not labeled in the challenge data, traditional segmentation losses (e.g., cross-entropy loss and Dice loss) can not be directly utilized for training. To enable effective training, we first group the data into different types considering their annotated label configurations. In our method, CT images with the same annotated classes are identified as the same type. During training, a certain labeling type is first selected with a probability proportional to its amount ratio, then a mini-batch of images from the selected type are randomly selected for partially supervised training. In this way, the annotated parts of images in a mini-batch are guaranteed to be the same.

*Merge-max operation* Mathematically, denote each input image as $\mathbf{I} \in \mathbb{R}^{Z \times H \times W}$, where $Z, H, W$ refers to the number of slices along each dimension. Then the generated logits can be denoted as $\mathbf{P} \in \mathbb{R}^{C \times Z \times H \times W}$, where $C$ is the total number of classes. Then $\mathbf{P}$ is split into labeled and unlabeled predictions $\mathbf{P}_l \in \mathbb{R}^{C_l \times Z \times H \times W}$ and $\mathbf{P}_u \in \mathbb{R}^{C_u \times Z \times H \times W}$, where $C_l + C_u = C$. $\mathbf{P}_l$ and $\mathbf{P}_u$ are further organized into a new prediction as,

$$\mathbf{P}' = [\mathbf{P}_l; max_{:C}(\mathbf{P}_u)], \tag{1}$$

where $[;]$ refers to concatenation along the first dimension, and $max_{:C}(\cdot)$ returns the max value along the $C$ dimension. The final reorganized predictions and groud truth labels are of size $(C_l + 1) \times Z \times H \times W$. Then traditional cross-entropy loss and Dice loss are calculated as the training loss. Instead of using the summation over all the unlabeled predictions, our proposed $merge - max$ operation can help to select the hardest unlabeled class for distinguishing between labeled classes in current mini-batch.

**Context-aware CutMix for online tumor augmentation** Another challenge in building a universal organ and tumor segmentation model is the lack of context-aware tumor cases, where a large fraction of the labeled tumor cases only contain tumors, while organs are labeled as background. To effectively take advantage of the labeled tumor cases, we propose a novel context-aware CutMix scheme for online tumor augmentation. Before that, we first generate pseudo labels for the cases that only contain labeled tumors using the fine model. Then Context-aware Tumor Objects (CTOs) are generated and processed for online tumor augmentation.

*Unlabeled data selection* Following Section 2.2, we can get the fine model trained on the partially labeled data with fine-resolution preprocessing. Then we make predictions and get pseudo labels for the unlabeled data. As the tumor prediction confidence is low ($<50\%$), we first select the most confident 1000 samples by sorting their organ prediction uncertainty. Then 100 out of the 1000 cases with the least tumor uncertainty are selected as the pseudo labeled samples for subsequent training. On the remaining 900 cases, the tumor class is relabeled as background for coarse model training. For details of uncertainty calculation, please refer to [9]. Please note, we separately calculate the uncertainty for organs and tumors. During the uncertainty calculation of organs, the pseudo labels generated by the FLARE22 winning algorithm [9] and the best-accuracy-algorithm [19] are utilized together with our fine model.

*Context-aware tumor objects (CTOs) generation* Since the shapes, positions, sizes and visual appearances of tumors may vary dramatically given the different disease types and progression in patients. Besides, for some cases, the tumors may be widely distributed in different parts of the body. It is not reasonable to use the tumor cases as a whole for online augmentation. To address this problem, we propose to generate context-aware tumor objects (CTOs), where each CTO contains only one connected tumor region. To further preserve the context of each tumor region, the neighboring regions with a predefined size are cropped together with the tumor object to form a context-aware tumor object. Figure 2 demonstrates some CTO slices on the transverse plane. In our paper, based on the statistics of tumor sizes, the neighboring extension sizes are randomly chosen between [8,16] and [24,32] along the $Z$ and $H, W$ dimension respectively. The cropped CTOs are processed following the same pre-processing steps as the coarse-resolution configurations, and saved offline for coarse model training.

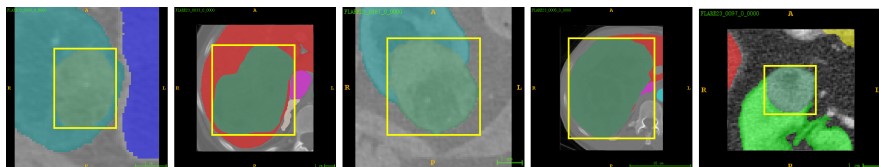

**Fig. 2.** Some sample slices of the generated CTO on the transverse plane. The middle-centered regions within each yellow bounding box are tumor objects. The neighboring areas are also cropped to capture context information.

*Online tumor augmentation* During the training of coarse model, for image cases with labeled tumors, they are directly optimized according to Section 2.2. As for images without labeled tumors, the offline generated CTOs are randomly selected and pasted into the abdominal regions of the current image for loss calculation. This is referred as the context-aware CutMix. Mathematically, denote the (image, label) pair of current image and the selected CTO as

$\mathbf{I} \in \mathbb{R}^{Z \times H \times W}, \mathbf{Y} \in \mathbb{R}^{Z \times H \times W}$ and $\mathbf{I}_{CTO} \in \mathbb{R}^{Z1 \times H1 \times W1}, \mathbf{Y}_{CTO} \in \mathbb{R}^{Z1 \times H1 \times W1}$ respectively, where $Z1, H1, W1$ are varying according to the size of the selected CTO. To perform context-aware CutMix, we first generate a zero-centered mask $\mathcal{M} \in \{0, 1\}^{Z \times H \times W}$, indicating whether the voxel comes from the CTO (0) or the current image (1). The size of the zero-value region is $Z1, H1, W1$. Please note the zero-center is constrained to be inside the abdominal region in our experiments. Formally, the image and label obtained using context-aware CutMix is formulated as,

$$\mathbf{I}_{CM} = \mathbf{I} \odot \mathcal{M} + \mathbf{I}_{CTO} \odot (1 - \mathcal{M}); \mathbf{Y}_{CM} = \mathbf{Y} \odot \mathcal{M} + \mathbf{Y}_{CTO} \odot (1 - \mathcal{M}), \quad (2)$$

where $\odot$ means element-wise multiplication. $\mathbf{I}_{CM}, \mathbf{Y}_{CM}$ are further utilized to train the coarse model according to Section 2.2.

*Data-cleaning for robust training* To further improve the training data quality, we introduce the data-cleaning strategy as follows. Firstly, we remove the very small isolated regions (100 voxels in our case), since they could be noise. Then we filter out the tumors that are not in the abdominal region. We further filter out tumor objects far away from all the organs to preserve context during training.

**Efficient adaptations to nnU-Net** To improve the inference efficiency, we make the following adaptations to the nnU-Net framework. We modify the cropping and resampling functions in the image preprocessing stage, where Pytorch interpolation is adopted for faster resampling. In [9], the authors assume that organs locate in the middle of the transverse plane in CT images. Inspired by this, we design a novel filtering strategy to filter out the sliding windows by only predicting a small fraction of the windows in the current transverse plane. In specific, when $2 \times 2$ ($3 \times 3$) windows exist in current transverse plane, we select the windows indexed as $0, 3$ (4) as the indicator. If the selected windows only contain the background, then the other windows within the same transverse plane is directly set as background. This strategy can effectively improve the inference efficiency, especially for large image cases.

## 3    Experiments and Results

### 3.1    Datasets and evaluation measures

The FLARE 2023 challenge is an extension of the FLARE 2021-2022 [13][14], aiming to promote the development of foundation models in abdominal disease analysis. The segmentation targets cover 13 organs and various abdominal lesions. The training dataset is curated from more than 30 medical centers under the license permission, including TCIA [2], LiTS [1], MSD [18], KiTS [6,7], autoPET [5,4], TotalSegmentator [20], and AbdomenCT-1K [15]. The training set includes 4000 abdomen CT scans where 2200 CT scans with partial labels and 1800 CT scans without labels. The validation and testing sets include 100 and 400 CT scans, respectively, which cover various abdominal cancer types, such as

liver cancer, kidney cancer, pancreas cancer, colon cancer, gastric cancer, and so on. The organ annotation process used ITK-SNAP [21], nnU-Net [10], and MedSAM [12].

The evaluation metrics encompass two accuracy measures—Dice Similarity Coefficient (DSC) and Normalized Surface Dice (NSD)—alongside two efficiency measures—running time and area under the GPU memory-time curve. These metrics collectively contribute to the ranking computation. Furthermore, the running time and GPU memory consumption are considered within tolerances of 15 seconds and 4 GB, respectively.

### 3.2   Implementation details

**Environment settings**  The development environments and requirements are presented in Table 1.

**Table 1.** Development environments and requirements.

| | |
|---|---|
| System | Ubuntu 20.04 LTS |
| CPU | Dual Intel Xeon Platinum 8168, 2.7GHz, 24 cores |
| RAM | 1.5TB |
| GPU (number and type) | 16x NVIDIA Tesla V100 |
| CUDA version | 11.7 |
| Programming language | Python 3.10 |
| Deep learning framework | torch 2.0, torchvision 0.15.1 |
| Specific dependencies | nnU-Net 2.0 |
| Code | https://github.com/luckieucas/FLARE23 |

**Table 2.** Training protocols.

| | |
|---|---|
| Network initialization | "He" normal initialization |
| Batch size | 4 |
| Patch size | 32×192×192 |
| Total epochs | 1200 |
| Optimizer | SGD with nesterov momentum($\mu = 0.99$) |
| Initial learning rate (lr) | 0.01 |
| Lr decay schedule | Poly learning rate policy:$(1 - epoch/1000)^{0.9}$ |
| Training time | 24 hours |
| Loss function | Dice loss and cross-entropy loss |
| Number of model parameters | 88.62M[3] |
| Number of flops | 2036G[4] |
| $CO_2$eq | 1 Kg[5] |

**Training protocols** During training, we first train a fine model on fine-resolution data (where the target spacing is set as (2.5, 0.82, 0.82)). Then, we generate pseudo labels for the unlabeled data using the trained fine model, and 1000 out of the 1800 unlabeled data are selected according to their organ prediction uncertainty. 100 out of the selected 1000 samples with least tumor uncertainty are further chosen as the pseudo labeled samples for training the coarse model (where the target spacing is set as (2.5, 1.5, 1.5)). While on the remaining 900 cases, online tumor augmentation is performed to enhance the tumor diversity during coarse model training. We adopt extensive data augmentations (including rotations, elastic deformations, and random cropping) to enhance our models' generalization capabilities. In our patch-based training, we over-sample the foreground classes with the oversampling percent set as 0.7. Our training batch size is set as 4. And the optimal models are selected based on their segmentation performance on the public validation set. More details of the training protocol are presented in Table 2.

## 4   Results and discussion

### 4.1   Quantitative results on validation set

Table 3 shows the quantitative results on validation set. Our method achieves a mean Dice Similarity Coefficient (DSC) of 88.90% and a Normalized Surface Dice (NSD) of 92.21% on the FLARE 2023 online validation dataset. Table 4 shows the effectiveness of using unlabeled data. One can see that by using unlabeled data for training, the segmentation performance improved.

**Table 3.** Quantitative evaluation results.

| Target | Public Validation | | Online Validation | | Testing | |
|---|---|---|---|---|---|---|
| | DSC(%) | NSD(%) | DSC(%) | NSD(%) | DSC(%) | NSD (%) |
| Liver | 98.40 ± 0.57 | 99.13 ± 0.99 | 98.39 | 99.15 | 97.26 | 98.17 |
| Right Kidney | 95.71 ± 8.65 | 95.86 ± 8.49 | 95.55 | 95.73 | 95.49 | 95.16 |
| Spleen | 98.37 ± 0.69 | 99.31 ± 1.20 | 98.47 | 99.51 | 97.71 | 98.81 |
| Pancreas | 88.13 ± 6.82 | 97.27 ± 4.79 | 87.50 | 96.91 | 92.03 | 97.99 |
| Aorta | 97.37 ± 1.64 | 99.13 ± 1.88 | 97.39 | 99.08 | 98.06 | 99.68 |
| Inferior vena cava | 96.01 ± 1.81 | 97.96 ± 2.36 | 95.97 | 97.86 | 96.63 | 98.70 |
| Right adrenal gland | 88.76 ± 2.99 | 97.78 ± 1.83 | 87.45 | 96.08 | 88.46 | 96.60 |
| Left adrenal gland | 88.09 ± 5.11 | 97.31 ± 3.58 | 88.01 | 96.58 | 89.59 | 97.15 |
| Gallbladder | 91.66 ± 7.88 | 92.26 ± 9.59 | 87.41 | 87.98 | 85.59 | 87.75 |
| Esophagus | 84.26 ± 14.91 | 92.90 ± 15.57 | 85.28 | 94.35 | 91.73 | 98.78 |
| Stomach | 93.89 ± 5.95 | 96.71 ± 6.57 | 94.74 | 97.52 | 95.00 | 97.81 |
| Duodenum | 86.20 ± 6.59 | 95.62 ± 4.53 | 86.87 | 96.13 | 90.99 | 97.82 |
| Left Kidney | 95.67 ± 5.67 | 95.07 ± 9.10 | 95.30 | 95.39 | 94.35 | 94.55 |
| Tumor | 52.76 ± 34.21 | 43.92 ± 31.68 | 46.26 | 38.65 | 61.49 | 49.90 |
| Average Organ | 92.50 ± 5.33 | 96.64 ± 5.42 | 92.18 | 96.33 | 93.17 | 96.76 |
| Average | 89.66 ± 7.39 | 92.87± 7.30 | 88.90 | 92.21 | 90.91 | 93.41 |

**Table 4.** The effect of using unlabeled data.

| Target | Train without unlabeled data | | Train with labeled and unlabeled data | |
|---|---|---|---|---|
| | DSC(%) | NSD(%) | DSC(%) | NSD(%) |
| Liver | 98.34 | 99.00 | 98.39 | 99.15 |
| Right Kidney | 95.20 | 95.84 | 95.55 | 95.73 |
| Spleen | 98.38 | 99.43 | 98.47 | 99.51 |
| Pancreas | 86.66 | 96.51 | 87.50 | 96.91 |
| Aorta | 96.99 | 98.61 | 97.39 | 99.08 |
| Inferior vena cava | 94.84 | 96.29 | 95.97 | 97.86 |
| Right adrenal gland | 87.97 | 97.35 | 87.45 | 96.08 |
| Left adrenal gland | 86.40 | 95.83 | 88.01 | 96.58 |
| Gallbladder | 84.43 | 84.86 | 87.41 | 87.98 |
| Esophagus | 83.72 | 93.08 | 85.28 | 94.35 |
| Stomach | 93.89 | 96.65 | 94.74 | 97.52 |
| Duodenum | 84.45 | 94.80 | 86.87 | 96.13 |
| Left kidney | 94.52 | 95.16 | 95.30 | 95.39 |
| Tumor | 42.65 | 34.90 | 46.26 | 38.65 |
| Average Organ | 91.20 | 95.65 | 92.18 | 96.33 |
| Average | 87.74 | 91.31 | 88.90 | 92.21 |

### 4.2   Qualitative results on validation set

We analyze the samples with relatively good predictions and those with poor predictions. Figure 3 shows the results. Cases #0017 and #0053 are good cases, it can be observed that the well-segmented cases have clear organ and tumor boundaries. Case #0067 and #0035 are bad cases, they often have poor predictions of results on tumors, this may due to the large differences in tumor size of different organs. Furthermore, when training with both labeled and unlabeled data, the segmentation results are more consistent with the ground truth compared to the segmentation results achieved by training with only labeled data.

### 4.3   Segmentation efficiency results on validation set

The average running time in validation set is 16.10 s per case in inference phase , and average used GPU memory is 2823 MB. The area under GPU memory-time curve is 18720. Table 5 lists segmentation efficiency of some typical cases.

### 4.4   Results on final testing set

We obtained scores of 93.17% (organ DSC), 96.76% (organ NSD), 61.49% (tumor DSC), and 49.9% (tumor NSD) on the official test set. The average time latency and memory usage on the test set were 13.95 seconds and 2823 MB, with an average area under the GPU memory-time curve of 14112. Collectively, we ranked second among all submitted teams.

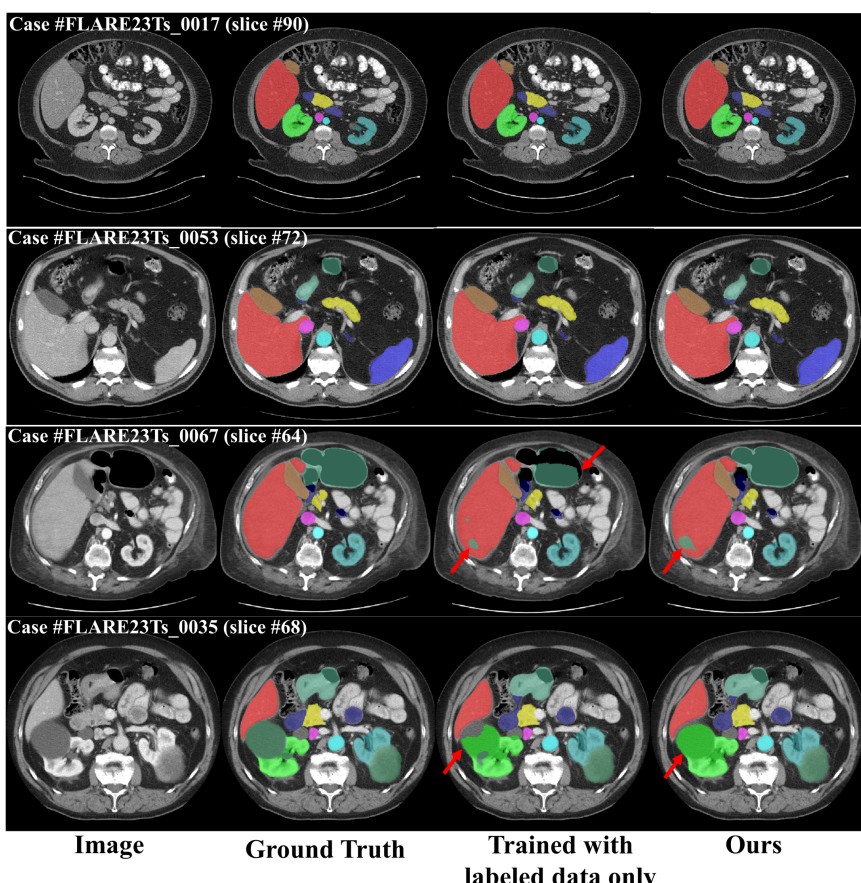

**Fig. 3.** Qualitative results on easy (case FLARETs #0017 and#0053) and hard (case FLARETs #0035 and #0067) examples. Please note that the results in the third column are generated by the model trained using only the partially labeled data.

**Table 5.** Quantitative evaluation of segmentation efficiency in terms of the running time and GPU memory consumption. Total GPU denotes the area under GPU Memory-Time curve. Evaluation GPU platform: NVIDIA QUADRO RTX5000 (16G).

| Case ID | Image Size | Running Time (s) | Max GPU (MB) | Total GPU (MB) |
|---------|-----------|------------------|--------------|----------------|
| 0001 | (512, 512, 55) | 15.02 | 2688 | 11549 |
| 0051 | (512, 512, 100) | 13.47 | 2962 | 14673 |
| 0017 | (512, 512, 150) | 24.04 | 3022 | 23779 |
| 0019 | (512, 512, 215) | 17.27 | 2810 | 19026 |
| 0099 | (512, 512, 334) | 19.24 | 2958 | 23661 |
| 0063 | (512, 512, 448) | 23.27 | 3020 | 30148 |
| 0048 | (512, 512, 499) | 24.74 | 2994 | 33037 |
| 0029 | (512, 512, 554) | 31.57 | 3184 | 43774 |

### 4.5   Discussion on unlabeled data

In our framework, the unlabeled data contributes from the following two aspects. Firstly, we select 100 pseudo labeled data with highest confidence to improve the data diversity, especially for tumors. Considering that the DSC performance of tumor is very low (less than 50%), only 100 out of the 1800 unlabeled samples are directly utilized to guarantee the pseudo labeling quality. Secondly, to take advantage of the confident organ parts from the unlabeled set, we select another 900 cases with best DSC performance on organs, and reset the tumor prediction as background. During training, the 900 cases will perform online CutMix with our generated CTOs to achieve dynamic tumor augmentation. In this way, we can effectively utilize the unlabeled data without introducing too much labeling noise.

### 4.6   Limitation and future work

We summarize the limitations of our method as follows: 1) Given that only 100 samples with tumors are utilized, the tumor cases in the unlabeled data is not fully exploited. 2) Due to different types of cancers, the non-rigidity of tumors and different disease progression across patients, the visual appearances of the pan-cancer class may vary dramatically among different individuals. Therefore, directly classifying the tumors as one unified class may be not the best choice. To address the first issue, we will resort to the noisy learning tricks to directly learn the knowledge from noisy pseudo labels. As for the second issue, we will try to incorporate more prior knowledge (e.g., typical positional correlation between each organ and the corresponding tumor) into our framework to enhance the tumor learning process.

## 5   Conclusion

In this paper, we propose a novel framework to achieve universal organ and tumor segmentation. Specifically, we introduce the merge-max operation to perform hard-mining on unlabeled classes. Furthermore, to properly utilize the labeled tumor cases, we propose a novel context-aware CutMix scheme. This online tumor augmentation strategy is demonstrated to boost the segmentation performance. We further validate that the data cleaning step is crucial to improve the NSD performance, especially for organs.

**Acknowledgements** The authors of this paper declare that the segmentation method they implemented for participation in the FLARE 2023 challenge has not used any pre-trained models nor additional datasets other than those provided by the organizers. The proposed solution is fully automatic without any manual intervention. We thank all the data owners for making the CT scans publicly available and CodaLab [16] for hosting the challenge platform.

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
