# OpenReview forum: "Context-aware Cutmix Is All You Need for Universal Organ and Cancer Segmentation"
_MICCAI.org/2023/FLARE — Submitted to FLARE 2023_

### Official Review · Reviewer_ZsJf · 2023-09-21
**Context-aware Cutmix Is All You Need for Universal Organ and Cancer Segmentation**

**Rating:** 7
**Confidence:** 5

**Review:**

Summary

This paper proposes a partially supervised learning method that incorporates modules of merge-max and CutMix.
The training of a robust segmentation network is achieved through data cleaning techniques.

Comments

1. DSC, NSD, Runtime and GPU memory results should be shown within the abstract.
2. Typographical errors, e.g. Fig. 3 occupies a full page.
3. The method in this paper achieves both excellent inference speed and segmentation accuracy, and it would be beneficial for the authors to provide open source code.

---

> ### Comment · Reviewer_ZsJf · 2023-11-20
> **2nd round Review**
>
> The new version of the paper is well structured.
>
> The proposed method in this paper achieves better segmentation results, and the effect of unlabeled data is analyzed in Table 4, but the Table 4 typography needs to be corrected.

---

### Official Review · Reviewer_shh4 · 2023-09-22
**Context-aware Cutmix Is All You Need for Universal Organ and Cancer Segmentation**

**Rating:** 7
**Confidence:** 4

**Review:**

Pros:

The method proposes a novel partially supervised segmentation framework for hard mining in unlabeled categories by introducing a merge-max operation. A novel context-aware CutMix scheme is designed for tumor enhancement. Finally, the efficiency is improved by tuning nnU-Net. Excellent segmentation results are obtained.

Cons:

Lack of ablation trials demonstrating the effectiveness of CutMix for tumor enhancement

---

> ### Comment · Reviewer_shh4 · 2023-11-23
> **Second round of evaluation**
>
> The paper is well structured, clearly explained, and the methodology is novel and close to publication standard.
>
> However, the formatting of Table 4 needs to be adjusted, and it is also puzzling that the ablation experiments of merge-max and cutmix have not been provided in the revised manuscript to demonstrate the degree of validity of the method.

---

### Official Review · Reviewer_bkQV · 2023-09-22
**Context-aware Cutmix Is All You Need for Universal Organ and Cancer Segmentation**

**Rating:** 7
**Confidence:** 5

**Review:**

Pros:
1. This paper proposes a novel partially supervised segmentation framework by introducing a merge−max operation for hard mining among the unlabeled classes.
2. Authors design a novel context-aware CutMix scheme to dynamically perform tumor augmentation during training.

Cons:
1. The DSC, NSD, Runtime, and GPU memory are not introduced in the abstract.

---

### Official Review · Reviewer_abjv · 2023-09-22
**Context-aware Cutmix Is All You Need for Universal Organ and Cancer Segmentation**

**Rating:** 7
**Confidence:** 4

**Review:**

This framework incorporates a merge-max operation for hard mining among the unlabeled classes, which helps improve the segmentation performance by effectively leveraging the available information from unlabeled data.

---

### Official Review · Reviewer_UfM5 · 2023-10-03
**Context-aware Cutmix Is All You Need for Universal Organ and Cancer Segmentation**

**Rating:** 7
**Confidence:** 5

**Review:**

This paper proposes a novel merge-max operation for hard mining among the unlabeled classes and CutMix scheme, which is designed to effectively enhance the tumor. A data-cleaning method is proposed to effectively improve the quality of training data.
Pros:
Clear structure, good writing.
Cons:
Lack of validation of merge-max and CutMix schemes.
The segmentation results and efficiency evaluations are not reflected in the abstract.

---

> ### Comment · Reviewer_UfM5 · 2023-11-30
> **Second Round Review**
>
> The revised manuscript exhibits a commendable structural organization.

---

### Official Review · Reviewer_m7zx · 2023-10-03
**Good paper but lack of some details**

**Rating:** 7
**Confidence:** 5

**Review:**

- Please add the performance of your proposed method on the online validation set in the asbstract, including organ and tumor DSC, organ and tumor NSD, running time, AUC of GPU memory.
- Did you use any other data augmentation except for tumor augmentation?

---

### Official Review · Reviewer_QTQc · 2023-10-04
**This paper is generally good. As many reviewer mentioned, more details are potentially needed.**

**Rating:** 7
**Confidence:** 5

**Review:**

As a challenge paper:
The manuscript needs details about experiments and how benchmark metrics are acquired. Including:
1. dataset usage in details, cross-validation, training time, model complexity, etc.
2. Are model ensemble required.
3. Time latency and memory usage as one of the significance in the FLARE challenge.

---

### Official Review · Reviewer_3tM8 · 2023-10-17
**Reviews**

**Rating:** 7
**Confidence:** 4

**Review:**

This paper presents a novel approach to tackle the challenges in universal organ and cancer segmentation, primarily focusing on issues related to partial and noisy labels and the high heterogeneity of tumor cases. The authors propose a partially supervised segmentation framework that incorporates a unique merge-max operation for hard mining among unlabeled classes and a context-aware CutMix scheme for dynamic tumor augmentation. Additionally, the paper suggests a data-cleaning strategy for self-training and adaptations to the nnU-Net framework to enhance efficiency.

The abstract can be enriched by adding a summary of the results obtained.

---

### Decision · Program_Chairs · 2023-10-24

Accept